# A phenotypic *Caenorhabditis elegans* screen identifies a selective suppressor of antipsychotic-induced hyperphagia

Anabel Perez-Gomez[1,2], Maria Carretero[1,2], Natalie Weber[3], Veronika Peterka[3], Alan To[1,2], Viktoriya Titova[1,2], Gregory Solis[1,2], Olivia Osborn[3] & Michael Petrascheck [1,2]

Antipsychotic (AP) drugs are used to treat psychiatric disorders but are associated with significant weight gain and metabolic disease. Increased food intake (hyperphagia) appears to be a driving force by which APs induce weight gain but the mechanisms are poorly understood. Here we report that administration of APs to *C. elegans* induces hyperphagia by a mechanism that is genetically distinct from basal food intake. We exploit this finding to screen for adjuvant drugs that suppress AP-induced hyperphagia in *C. elegans* and mice. In mice AP-induced hyperphagia is associated with a unique hypothalamic gene expression signature that is abrogated by adjuvant drug treatment. Genetic analysis of this signature using *C. elegans* identifies two transcription factors, *nhr-25*/Nr5a2 and *nfyb-1*/NFYB to be required for AP-induced hyperphagia. Our study reveals that AP-induced hyperphagia can be selectively suppressed without affecting basal food intake allowing for novel drug discovery strategies to combat AP-induced metabolic side effects.

[1] Department of Molecular Medicine, The Scripps Research Institute, 10550 North Torrey Pines Road, La Jolla, CA 92037, USA. [2] Department of Neuroscience, The Scripps Research Institute, 10550 North Torrey Pines Road, La Jolla, CA 92037, USA. [3] Department of Medicine, University of California, San Diego, 9500 Gilman Drive, La Jolla, CA 92093, USA. These authors contributed equally: Anabel Perez-Gomez, Maria Carretero. These authors jointly supervised this work: Olivia Osborn, Michael Petrascheck. Correspondence and requests for materials should be addressed to O.O. (email: oosborn@ucsd.edu) or to M.P. (email: pscheck@scripps.edu)

Many commonly prescribed medications have the unwanted side effect of weight gain[1–3]. While the magnitude of the effect varies depending on the drug and between individuals, antipsychotic (AP) drugs are associated with some of the most severe effects on weight gain[4]. Approximately 4 million people in the USA are currently prescribed AP medications to manage a range of illnesses such as schizophrenia and bipolar disorder[2,5,6]. More recently, the use of APs has been extended to cover a wide range of psychiatric conditions including off-label use in depression, anxiety, dementia, insomnia and, post traumatic stress disorder[6,7]. While first generation APs, discovered in the 1950s, revolutionized psychiatric healthcare by treating hallucinations and delusions, they were also associated with unwanted side effects including movement disorders and weight gain[8]. Second generation APs are more effective in treating the complex aspects of psychiatric condition and cause over-all less side effects. However, treatment with second generation APs leads to an even greater weight gain compared to first generation APs along with a high risk of development of associated metabolic disease[9,10]. Notably, the incidence of diabetes among second generation AP users is four times higher than age-matched, race-matched, and sex-matched controls[2]. Metabolic side effects are also the most commonly reported reason for non-compliance with second generation AP medication[11]. Meta-analysis of the weight gain potential of various APs drugs revealed that 41% of patients taking olanzapine experienced clinically significant increase in body weight compared with 20% for clozapine or 23% for quetiapine[12]. While olanzapine has the highest risk for weight gain, it is also regarded as one of the most clinically effective medications[13] but this efficacy at treatment of psychosis must always be balanced with the likelihood of weight gain and subsequent metabolic side effects.

Excessive food intake, termed hyperphagia, appears to be a driving force by which APs induce weight gain[1,14,15]. Studies have found that APs are associated with increased appetite and food craving[15–17], especially for sweet and fatty highly palatable foods[18]. Feeding behavior is a highly complex process involving signaling between peripheral tissues and the hypothalamus, integrating signals in the central nervous system. This complex regulation has made identifying the mechanism by which APs induce aberrant feeding behavior particularly difficult. Correlative pharmacological studies determining binding profiles of APs suggest some of their weight gain effects may be driven through their complex serotonergic, histaminergic and dopaminergic receptor binding activities[19]. However, there is still a general lack of understanding of the specific mechanisms underlying their potent metabolic side effects[20].

Feeding is key to survival, and while nutritional demands vary considerably across different metazoans, they all rely on essentially the same macronutrients such as carbohydrates, fats and amino acids to satisfy energy demands. Therefore, it is likely that many aspects of feeding are regulated by core ancestral mechanisms. AP-induced hyperphagia is a complex behavior involving both physiological and behavioral changes that cannot be replicated in cell culture systems and require a whole animal model. Pre-clinical rodent models have been established that replicate the weight gain effects of AP-treated human patients but are not suited for a screening based approach[21].

Here we report that administration of APs to the small nematode C. elegans induces hyperphagia by a mechanism that is genetically distinct from normal, unstimulated food intake, termed basal feeding. We exploit this observation to screen 192 drugs for those that selectively suppress AP-induced hyperphagia. Follow up testing in mice shows that minocycline, an adjuvant drug identified in the C. elegans screen, blocks AP-induced hyperphagia, weight gain, and AP-induced hypothalamic gene expression in mice. As in C. elegans, the mouse data reveal the existence of a distinct hyperphagic mechanism that can be selectively suppressed without affecting basal feeding. Minocycline has also shown efficacy in reducing weight gain in response to AP treatment in clinical studies where only 40% of the co-treated group gained any weight at all compared with 100% in the control group[22]. Our discovery based and hypothesis-free C. elegans screening strategy facilitates the identification of other potential adjuvant options, as well as gain deeper mechanistic understanding in to the mode of action of effective adjuvants like minocycline.

## Results

**APs induce hyperphagia in *Caenorhabditis elegans*.** C. elegans eat bacteria and feeding can be accurately determined by measuring the change in bacterial concentration over time (Fig. 1a). Using our C. elegans based 96-well plate food-intake assay[23], we determined that various classes of drugs, including antihistamines, tricyclic antidepressants, and APs, known to induce hyperphagia in human patients[10,24], also result in significantly increased food intake in C. elegans (Table 1). We tested a variety of APs (including first and second generation classes) known to induce weight gain to various degrees in human patients[4] and measured the effect of these drugs on food intake in C. elegans. Remarkably, 9 out of 12 (75%) APs tested induced hyperphagia in C. elegans (Table 1). We determined $EC_{50}$ values in vivo for the APs that induced hyperphagia in C. elegans obtaining values ranging from ~40 nM for the newest APs asenapine and ziprasidone to ~36 μM for the AP olanzapine (Fig. 1b, Table 1). Of note, APs induced hyperphagia in C. elegans even when fed γ-irradiated (dead) bacteria suggesting the AP-induced hyperphagic effect was not due to an interaction with the bacteria. Taken together, these results show that the hyperphagic side effects of many psychotropic drugs are also observed in C. elegans. These data suggest that C. elegans can be used as a model organism to investigate the mechanisms behind AP-induced changes in feeding behavior.

**AP-induced hyperphagia is distinct from basal food intake.** Due to the established role of dopamine[25] and serotonin[26] in the regulation of appetite, it has been speculated that weight gain may be an on-target side-effect inextricably linked to the therapeutic effect of APs. We used C. elegans strains with mutations in dopaminergic and serotonergic receptors to determine the role of these signaling pathways in AP-induced hyperphagia.

AP treatment in all single dopamine receptor mutants resulted in a hyperphagic response similar to that of wild type (N2) animals (Fig. 1c). To account for the possibility of redundant receptor action we also assayed triple and quadruple dopamine receptor mutants for their response to APs. Untreated triple and quadruple dopamine receptor mutants ate less than wild type N2 animals, confirming the role of dopamine in basal food intake, but responded to APs similar to wild type animals (Fig. 1d). Thus, dopaminergic receptors play a role in the regulation of basal food intake, but are dispensable for the AP-induced hyperphagic effect.

While AP treatment in single serotonin receptor mutants *ser-1 (ok345)*, *ser-4(ok512)*, and *ser-7(tm1325)* elicited a similar hyperphagic effect observed in wild type (N2) animals (Fig. 1e), the *ser-5(ok3087)* mutant showed a severely blunted hyperphagic response. We then ablated the entire *ser-5* coding region to generate *ser-5(vq1)* knockout worms. These animals also did not elicit a hyperphagic response to AP treatment (Fig. 1f). To determine if these *ser-5(ok3087)* and *ser-5(vq1)* mutant animals have a general feeding defect or if they remain capable of mounting a hyperphagic response, we measured their food intake

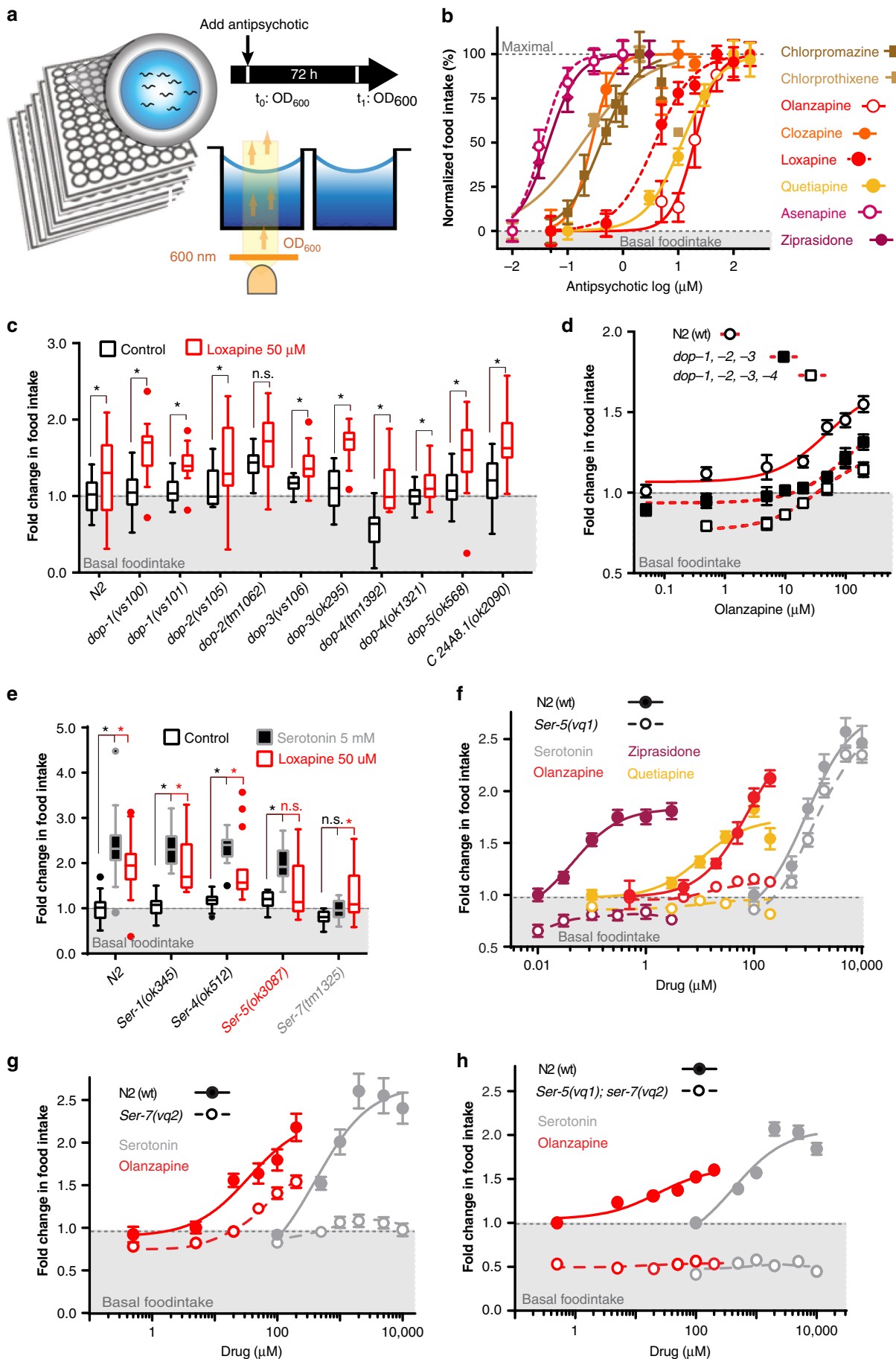

**Fig. 1** AP administration results in hyperphagic response in *C. elegans*. **a** Schematic diagram of *C. elegans* based food intake assay. **b** EC$_{50}$s of effect of various APs on food intake in *C. elegans*. **c** Food intake response of various *C. elegans* strains with single mutations or **d** triple/quadruple mutations in dopamine and serotonin signaling pathways compared with N2 (wild type) worms. **e** Comparison of food intake response in serotonin receptor mutants (*ser-1, ser-4, ser-5,* and *ser-7*) after either AP administration or serotonin administration. **f** Food intake response in a Crispr/Cas9 knock out strain *ser-5(vq1)* or N2 worms after treatment with various APs (olanzapine, ziprasidone, quetiapine) or serotonin as a hyperphagic control stimulus. **g** Food intake response in a Crispr/Cas9 knock out strain *ser-7(vq2)* or N2 worms after treatment with olanzapine or serotonin as a hyperphagic control stimulus. **h** Food intake response of *ser-5(vq1); ser-7(vq2)* double knockout or N2 worms after treatment olanzapine or serotonin as a hyperphagic control stimulus. Error bars in dose response plots (**b, d, f, g, h**) represent +/− S.E.M of 14-21 replicate wells with 5–15 animals each. **c** and **e** are Tukey style plots with the box representing the upper and lower quartile. Asterisks (*) represent $p < 0.05$ (ANOVA, Holm-Sidak corrected), n.s., not significant. Data (**c–f**) expressed as mean fold change in food intake relative to food intake of untreated N2 (wild-type) *C. elegans* assayed in parallel. For number of animals and P values see Supplementary Data 2. Source data are provided as a Source Data file

**Table 1 Comparison of weight gain side-effect of various drugs in human patients with effect on food intake in *C. elegans***

| Class | Drug | Brand name | Weight gain in humans | Effect on food intake in *C. elegans* | In vivo EC$_{50}$ μM | Ref. |
|---|---|---|---|---|---|---|
| Antihistamine | Terfenadine | Seldane | Intermediate | Hyperphagia | N.D | 2, 19 |
| Antihistamine | Cyproheptadine | Periactin | Intermediate | Hyperphagia | N.D | 2, 19 |
| Antidepressant | Mianserin | Tolvon | Intermediate | Hyperphagia | N.D | 1–3 |
| Antidepressant | Amitriptyline | Elavil | Intermediate | Hyperphagia | N.D | 1–3, 8 |
| 1st generation AP | Chlorpromazine | Thorazine | Substantial | Hyperphagia | 0.43 | 4, 8, 9, 12, 19 |
| 1st generation AP | Chlorprothixene | Taractan | no reports found | Hyperphagia | N.D | N/A |
| 1st generation AP | Flupenthixol | Depixol/ Fluanxol | low | Hyperphagia | N.D. | 4, 12 |
| 1st generation AP | Haloperidol | Haldol | low | No effect | N.A. | 4, 8, 9, 12, 19 |
| 1st generation AP | Loxapine | Loxitane | low | Hyperphagia | 5.7 | 4, 8 |
| 2nd generation AP | Clozapine | Clozaril | substantial | Hyperphagia | 2 | 1, 2, 4, 8, 9, 12, 19 |
| 2nd generation AP | Olanzapine | Zyprexa | Substantial | Hyperphagia | 36.9 | 1, 2, 4, 8, 9, 12, 19 |
| 2nd generation AP | Quetiapine | Seroquel | Intermediate | Hyperphagia | 11.6 | 4, 8, 9, 12, 19 |
| 2nd generation AP | Risperidone | Risperdal | Intermediate | No effect | N.A | 1, 4, 8, 9, 12 |
| 2nd generation AP | Lurasidone | Latuda | Neutral/low | No effect | N.A. | 4, 8, 9, 19 |
| 2nd generation AP | Asenapine | Saphris/Sycrest | Low | Hyperphagia | 0.03 | 4, 8, 9, 19 |
| 3rd generation AP | Aripiprazole | Abilify | Neutral/low | No effect | N.A. | 2, 4, 8, 9, 12, 19 |
| 3rd generation AP | Ziprasidone | Geodon | Neutral/low | Hyperphagia | 0.05 | 1, 2, 4, 8, 9, 12, 19 |

response to serotonin treatment, a potent hyperphagic signal in *C. elegans*[27]. Their hyperphagic response to serotonin was indistinguishable from wild type animals, showing this receptor to be specifically required for AP-induced hyperphagia (Fig. 1e, f). Interestingly, *ser-7* mutants elicited a hyperphagic response to AP treatment but did not respond to serotonin (Fig. 1e). We also generated a *ser-7(vq2)* knock out strain and tested its response to both serotonin and APs. The *ser-7(vq2)* strain did not respond to serotonin, but treatment with APs resulted in a strong hyperphagic response, albeit slightly less than in the N2 strain (Fig. 1g). We also generated a double knockout *ser-5(vq1); ser-7 (vq2)* strain that, as expected, did not respond to either serotonin or APs (Fig. 1h). These genetic data show that in *C. elegans*, AP-induced hyperphagia is based on a specific mechanism that is distinct from basal feeding. To determine if APs induce a similar, selective hyperphagic mechanism in mammals, we decided to screen *C. elegans* for a selective suppressor drug of AP-induced hyperphagia that then can be tested in mice.

**A screen for chemical suppressors of AP-induced hyperphagia.** An ideal adjuvant drug would block the hyperphagic and weight gain effects of APs without affecting basal food intake. To provide a rapid potential path to clinical translation we chose to start by screening FDA approved drugs for their ability to suppress AP-induced hyperphagia. We screened 192 drugs in combination with the AP loxapine and measured food intake after 72 h (Fig. 2a, Supplementary Data 1). We chose loxapine to induce

hyperphagia because it is one of the few water soluble APs and therefore allows minimizing the DMSO concentration. Minimizing DMSO proved critical for the screen as DMSO concentrations above 0.7% affect *C. elegans* feeding (data not shown). Each drug was ranked based on its ability to suppress AP-induced hyperphagia (Fig. 2b). We excluded drugs that directly lysed bacteria in the absence of worms, those with established anthelmintic activity or those that caused the worms to phenotypically appear sick (thin, immobile), indicative of drug toxicity. This primary screen identified 5 adjuvant candidate drugs, theophylline, idoxuridine, vemurafenib, doxycycline and scopine to suppress AP-induced hyperphagia (Fig. 2b). For each of these 5 primary hits we ordered a new independently synthesized sample and re-tested the effect of the drug on feeding behavior in *C. elegans*. Doxycycline, scopine and vemurafenib all suppressed AP-induced hyperphagia while new samples of theophylline and idoxuridine did not replicate the effect observed in the screen (data not shown), suggesting they were false positives. Alternatively, the abrogation of AP-induced hyperphagia by the original samples of theophylline and idoxuridine could be the result of a degradation product in the library rather than the original compound.

The identification of a positive hit for a tetracyclic antibiotic (doxycycline) in our screen prompted us to re-consider the possibility that the effect on AP-induced hyperphagia was due to antibiotic activity, despite the use of γ-irradiated (dead) bacteria in the screen. We therefore tested eight structurally related tetracycline antibiotics and two other classes of antibiotics

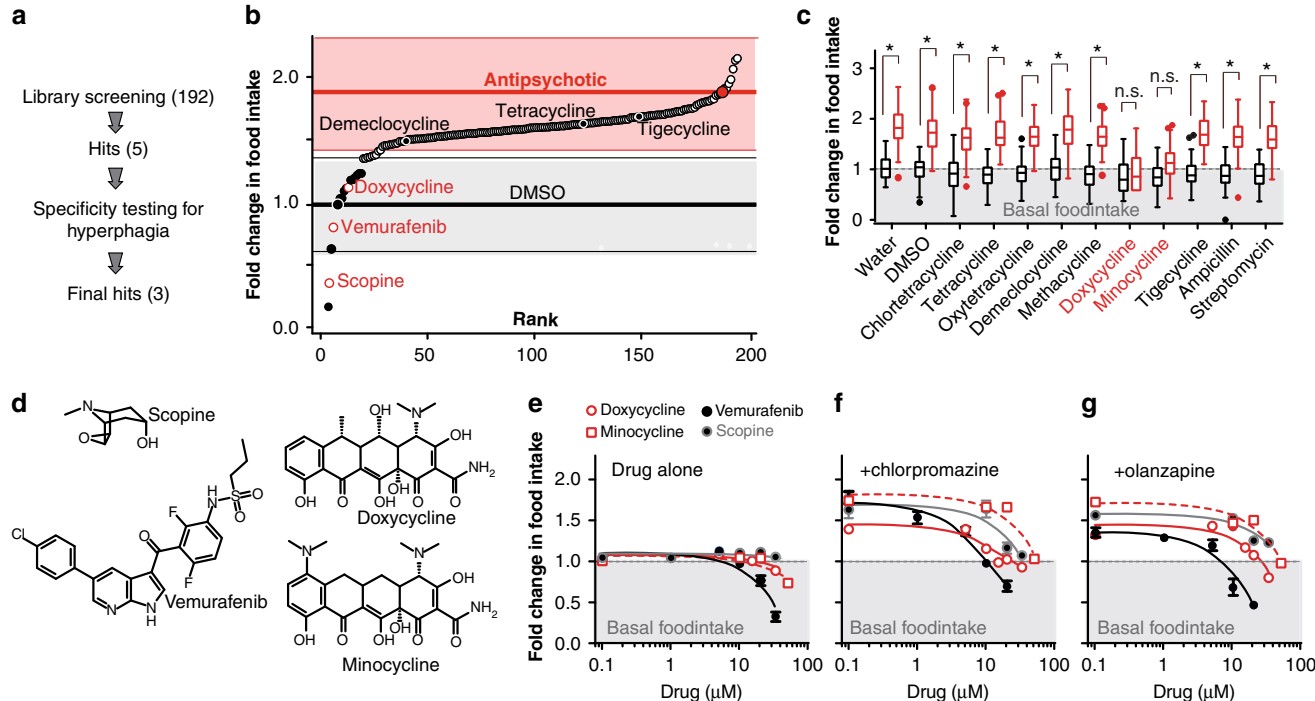

**Fig. 2** *C. elegans* based screen of FDA-approved drugs for potential adjuvant to block AP-induced hyperphagia. **a** Diagram of screening process. **b** Food intake based screen of 192 compounds in *C. elegans* in the presence of 50 μM of the AP loxapine (See Supplementary Data 1 and 2 for details). **c** Tukey style plot representing effects on food intake of various antibiotics alone (33 μM, black) or in combination with the AP loxapine (50 μM, red). **d** Structural comparison of primary hits identified from *C. elegans* based food intake screen. **e** Effect of primary hits on basal food intake in the absence of AP. **f** Effect of primary hits on chlorpromazine-induced hyperphagia. **g** Effect of primary hits on olanzapine-induced hyperphagia. Error bars in **e**–**g** represent +/− S.E.M. Food intake was measured using the bacterial clearance assay,[23] feeding dead, γ-irradiated bacteria. Data expressed as mean fold change in food intake relative to untreated N2 (wild-type) after 72 h. Asterisks (*) represent $p < 0.05$ (ANOVA, Bonferroni corrected), n.s. not significant. Source data are provided as a Source Data file

(ampicillin and streptomycin) for their ability to suppress AP-induced hyperphagia. Only doxycycline and minocycline blocked AP-induced hyperphagia, while six other structurally related tetracyclines did not suppress feeding, indicating a mechanism unrelated to their antibiotic properties (Fig. 2c).

We next tested whether these potential adjuvant candidates (doxycycline, minocycline, scopine, and vemurafenib, Fig. 2d) had any effect on basal feeding. Vemurafenib, an anticancer drug, decreased basal food intake, suggesting a non-specific effect on all forms of feeding, while minocycline, doxycycline, and scopine showed no effect on basal feeding unless used at high concentrations (Fig. 2e). We then tested if their abrogation of AP-induced hyperphagia was specific to loxapine or applicable to broader classes of APs including chlorpromazine (Fig. 2f) and olanzapine (Fig. 2g). All four drugs were efficacious in blocking hyperphagia induced by chlorpromazine (Fig. 2f) while scopine appeared to be less effective than the other three drugs in suppressing olanzapine-induced hyperphagia (Fig. 2g). Taken together, these data suggested that minocycline and doxycycline were the strongest adjuvant candidates as they had little effect on basal feeding behavior and suppressed AP-induced hyperphagia across a range of APs tested.

Minocycline is approved for the treatment of acne but was found to also exert anti-inflammatory, anti-oxidant and anti-apoptotic effects independent from its anti-microbial activity[28]. Minocycline crosses the blood brain barrier[29] and has been investigated in many neurological diseases[30,31] including psychiatric disease, with some, but not all, studies noting a reduction in weight gain (see discussion)[32]. Based on the superior safety profile and a lower HMIS hazard rating for minocycline

compared to doxycycline (HMIS: hazardous material identification system), as well as the literature suggesting a therapeutic potential for minocycline in psychiatric disease, we selected minocycline as the top candidate to evaluate its ability to suppress AP-induced hyperphagia in mice.

**Minocycline suppresses AP-induced hyperphagia in mice.** To test the ability of minocycline to suppress AP-induced hyperphagia in mammals we used a well-established mouse model to replicate AP-induced hyperphagia and weight gain[33,34]. This experimental mouse model incorporates olanzapine (54 mg/ kg) into a high fat diet that is fed to female *C57BL/6* mice[34]. In this model, co-treatment of minocycline suppressed olanzapine-induced hyperphagia and body weight gain after 1 week of feeding (Fig. 3a, b) and persisted throughout the chronic study (Fig. 3c). EchoMRI measurement of body composition showed that co-treatment with minocycline significantly suppressed the olanzapine-induced increase in fat mass (Fig. 3d), while lean mass remained unchanged between treatment groups (Fig. 3e). Importantly, minocycline treatment alone had no effect on weight gain and hyperphagia in the absence of olanzapine. Thus, minocycline did not act as a general weight loss drug, but selectively suppressed olanzapine induced hyperphagia and weight gain without affecting other homeostatic feeding mechanisms. While closely related antibiotics to minocycline had no effect on AP-induced food intake in worms we also tested for their potential to abrogate AP-induced food intake in mice. Co-treatment of tetracycline with olanzapine (TETRA + OLZ) did not blunt olanzapine-induced food intake or weight gain (Supplementary Fig. 2A-C). Therefore, we conclude that

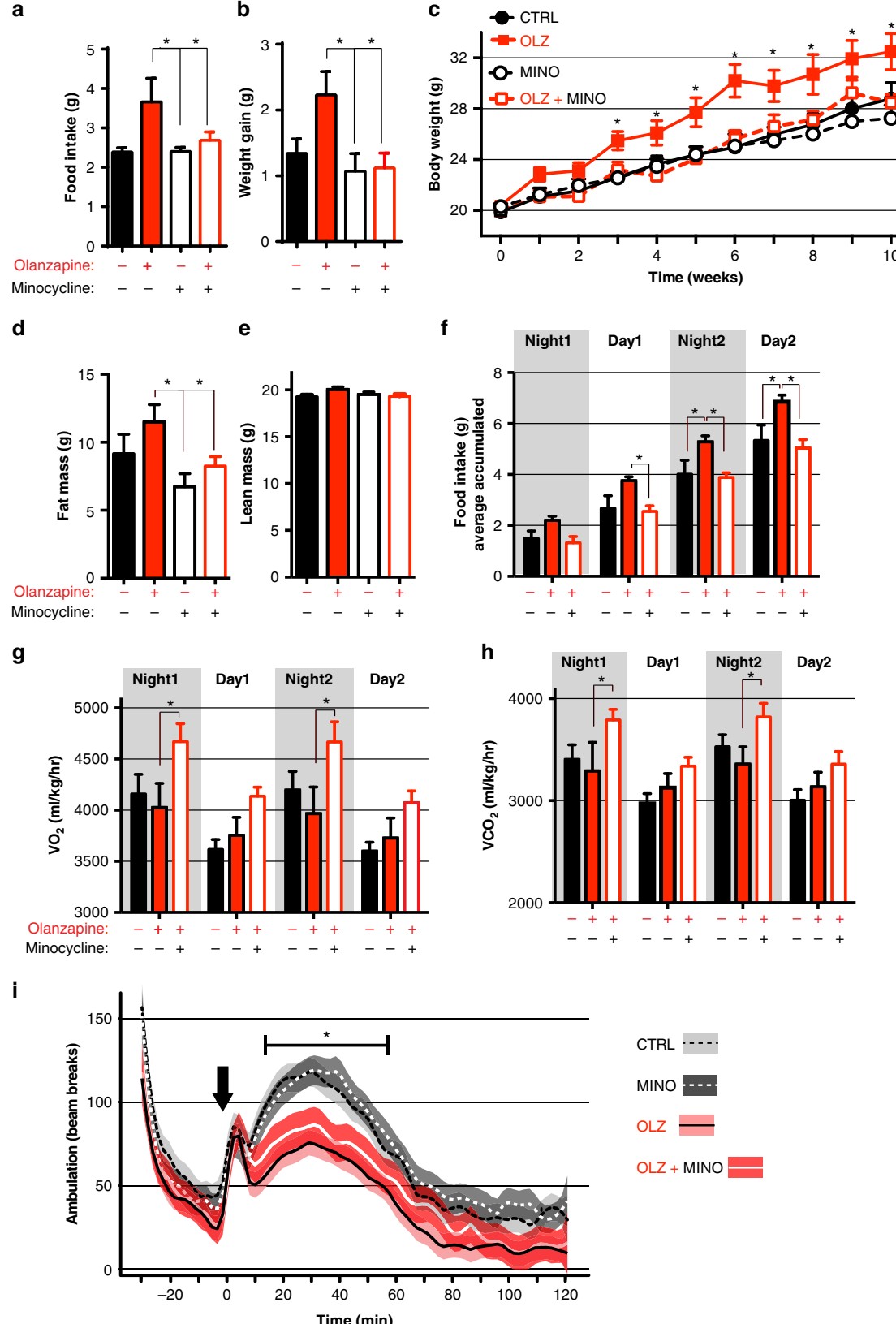

minocycline suppression of AP-induced food intake and weight gain is via a mechanism distinct from its antibiotic properties as closely related antibiotics do not suppress AP-induced food intake or weight gain.

To characterize the long term effects of olanzapine treatment and their supression by minocycline co-treatment, mice were transferred to metabolic chambers at week 11 of our study. At this time, minocycline co-treatement still suppressed the hyperphagic

**Fig. 3** Co-administration of minocycline in mice prevents olanzapine-induced hyperphagia and weight gain. **a** Food intake and (**b**) Body weight after 1 week of treatment. **c** Chronic administration of minocycline (MINO) prevents olanzapine-induced (OLZ) weight gain. **d, e** EchoMRI measurements of fat mass (**d**) and lean mass (**e**). **f–h** Metabolic chamber analysis of food intake (**f**), and oxygen consumption (VO$_2$, **g**) and carbon dioxide production (VCO$_2$, **h**). **i** Plots locomotion as a function of time before and after amphetamine administration (black arrow). Amphetamine induced a significant increase in locomotor activity that was attenuated by OLZ treatment. Co-treatment with MINO did not affect the ability of OLZ to suppress amphetamine-induced locomotor activity. For **a–f**, data are presented as mean per group +/− S.E.M. **a–e**. $n = 10$ per group, **f–h**. $n = 4$ per group in metabolic chambers, **i** $n = 12$ per group. **a–b, d–e** *$p < 0.05$, One-way ANOVA with uncorrected Fishers LSD test. **c, f–h, i** *$p < 0.05$ indicates significance using Two-way repeated measures ANOVA with Tukey multiple comparisons test. **i** Shaded background indicates 95% confidence interval or each cohort. Source data are provided as a Source Data file

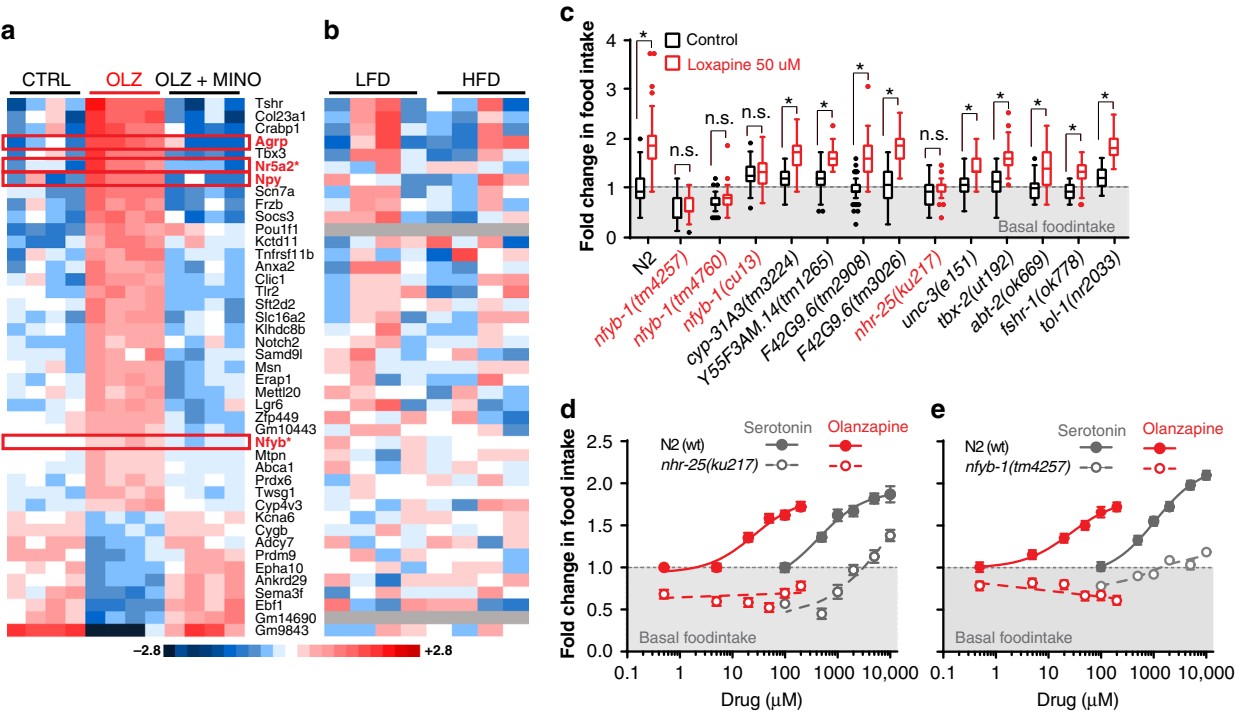

**Fig. 4** RNA sequencing identifies gene expression profile associated with AP-hyperphagia. **a** RNA-seq of hypothalamus of mice identifies unique signature of genes associated with AP-induced hyperphagia, $n = 4$ mice per group, control (CTRL), olanzapine (OLZ), and olanzapine + minocycline (OLZ + MINO). **b** Comparison of AP-induced profile genes in model of low fat diet (LFD) and high fat diet (HFD) fed mice ($n = 4$ per group). C. Testing AP-induced hyperphagia in various orthologous *C. elegans* mutants of genes identified from mouse hypothalamic RNA seq. **d, e** Comparison of food intake response of *nhr-25* mutant (**d**) and (**e**) *nfyb-1* mutant strains, to olanzapine induced hyperphagia and serotonin-induced hyperphagia compared with N2 (wild type control worms). Error bars in dose response plots (**d, e**) represent mean and +/− S.E.M of 14–21 replicate wells with 5–15 animals each. Data in **c** are represented by a tukey style plots with the box representing the upper and lower quartile. ANOVA, Holm-Sidak corrected. For number of animals see Supplementary Data 2

effect of olanzapine (Fig. 3f). In addition, minocycline co-treatment increased oxygen consumption and carbon dioxide production (Fig. 3g, h) indicating increased energy expenditure compared with olanzapine treatment alone. As observed in other rodent studies, olanzapine treatment results in decreased activity[35,36], however minocycline treatment did not have any additional effect on activity levels (Supplementary Fig. 1A-B). There were no differences in respiratory exchange ratio (RER) (Supplementary Fig. 1C) or levels of inflammatory cytokines or gut-derived peptides between treatment groups (Supplementary Fig. 1D).

It has been previously speculated that the metabolic side-effects may be inextricably linked to the therapeutic effect of APs[37]. To determine if co-treatment with minocycline impaired the therapeutic efficacy of olanzapine we used a standard test for antipsychotic efficacy in mice (amphetamine-induced hyperloco-motion)[38–40]. As expected, amphetamine treatment resulted in hyperactivity that was blunted by olanzapine treatment (Fig. 3i). Importantly, co-treatment with minocycline did not reduce the

ability of olanzapine to supress amphetamine-induced hyperlo-comotion (Fig. 3i), suggesting minocycline specifically supresses the metabolic side effects of olanzapine without affecting antipsychotic efficacy.

**An AP-induced hypothalamic gene expression signature.** In mammals the hypothalamus integrates peripheral metabolic signals to regulate feeding behavior. To gain mechanistic insights into how minocycline blocks AP-induced hyperphagia we determined gene expression changes in the hypothalamus by RNA-seq in control mice (no drug treatment), mice treated with olanzapine alone, or in mice co-treated with olanzapine and minocycline. We observed the specific regulation of 43 hypothalamic transcripts that were differentially expressed by olanzapine treatment compared with control and "corrected" in olanzapine treated mice that were co-treated with minocycline (FDR less than 0.3, Fig. 4a). Within this signature, 33/43 genes

were induced by olanzapine treatment and 10/43 genes were decreased by olanzapine treatment.

Two key orexigenic peptides (pro-feeding) NPY and AgRP were strongly induced in the hypothalamic gene expression signature upon olanzapine treatment[41,42](Fig. 4a). The induction of NPY[43–47] and AgRP[45] has previously been associated with AP treatment. Our studies show for the first time that suppression of the AP induced metabolic side effects by minocycline co-treatment also blocks NPY and AgRP induction, supporting the notion that these orexigenic peptides play a key role in AP-induced hyperphagia.

Given the general role of NPY and AgRP in the regulation of feeding, we also considered the possibility that the effect of minocycline on the hypothalamic gene expression could be a secondary effect on gene expression caused by differences in body weight and adiposity. To test this possibility we compared hypothalamic gene expression changes between normal and obese mice that were fed a low fat diet or a high fat diet, respectively. None of these genes showed differential expression between chronically fed high fat diet-obese mice compared with low fat diet-fed control mice (Fig. 4b). Thus, the hypothalamic gene expression signature is not caused by differences in weight, but the result of olanzapine treatment.

An unexplained phenomenon in rodent models of AP-induced weigh gain is that only females gain weight when treated with APs[48–50]. We exploited the lack of an olanzapine-induced weight gain in male mice to determine if the olanzapine-induced hypothalamic gene expression signature is specifically associated with hyperphagia and weight gain or a general signature associated with olanzapine treatment, irrespective of a metabolic phenotype. Quantitative PCR revealed that none of the 43 hypothalamic signature genes changed expression in male mice treated with olanzapine (Supplementary Fig. 3). These results show that olanzapine treatment leads to the induction of a hypothalamic gene expression signature that is closely associated with hyperphagia and weight gain. This raised the question whether some of the observed gene expression changes mediate AP-induced hyperphagia and weight gain.

**Hypothalamic signature genes mediate AP-induced hyperphagia.** To determine which genes in the AP-induced hypothalamic gene expression signature identified in mice are directly responsible for AP-induced hyperphagia, we returned to the *C. elegans* model system. We were unable to unequivocally assign all genes in the AP-induced hypothalamic gene expression signature to a single *C. elegans* ortholog. For example, while *C. elegans* has many conserved neuropeptide signaling systems that regulate metabolic pathways involved in appetite, satiety, energy homeostasis, and fat metabolism,[51] the precise orthologs of NPY or AgRP in *C. elegans* are currently unclear[52–54]. We tested 13 strains carrying mutations in the most obvious orthologs and quantified their food intake in response to the AP loxapine (Fig. 4c). Mutations in the transcription factor *nfyb-1*/Nfyb and the nuclear hormone receptor *nhr-25*/Nr5a2 abrogated loxapine-induced hyperphagia. Mutations in both transcription factors also abrogated hyperphagia induced by olanzapine, but not by serotonin, confirming the essential role of *nfyb-1*/Nfyb and *nhr-25*/Nr5a2 in AP-induced hyperphagia (Fig. 4d, e). Our results also suggested slightly different roles for the two factors. The *nfyb-1* (tm4257) mutant showed a very mild effect on basal food intake, but had a severely blunted response to serotonin. Thus, *nfyb-1* appears to be generally required for hyperphagia. The *nhr-25* (ku217) mutant showed a serotonin response profile similar to wild-type but its basal food intake was clearly reduced (Fig. 4d, e). The mammalian ortholog of *nhr-25*, nuclear receptor subfamily 5

group A member 2 (Nra5a2) has a well established role in liver lipid homeostasis[55–57] and pancreatic inflammation[58]. While NR5a2 is expressed in the hypothalamus[59,60], and plays a role in reproduction[61], its role in feeding behavior has not yet been described. Nfyb is part of the trimeric NF-Y complex previously shown to activate metabolic pathways[62] in cancer cells and to regulate transcription of leptin but has no described role in feeding behavior[63]. Taken together these data suggest *nhr-25* and *nfyb-1* to be required for AP-induced hyperphagia in *C. elegans*, and suggest that the induction of their mouse homologs Nra5a2 and Nfyb in the hypothalamus may play a role in AP-induced hyperphagia in mice.

In summary, our work identifies *C. elegans* as a screening model to identify candidate drugs that could be used to develop adjuvant treatments to abrogate AP-induced hyperphagia and weight gain in patients. As a proof of principle we identify minocycline as a selective suppressor of AP-induced metabolic phenotypes that include hyperphagia, weight gain, adiposity and the induction of a hypothalamic gene expression signature that was specifically associated with the hyperphagic phenotype in mice. Our data provide multiple lines of evidence, in two very different species, that AP induced hyperphagia is distinctly regulated from basal feeding and that it can be selectively suppressed.

## Discussion

Using our microtiter-based food intake assay[23] we report the surprising observation that AP-induced hyperphagic side effects widely observed in humans, are also observed in the basic organism *C. elegans*. Currently, no other model organism is known to replicate the AP-induced hyperphagia observed in human to such a remarkable extent for so many different APs. Our data suggests that AP-induced hyperphagia is distinct from basal feeding mechanisms and thus it can be pharmacologically blocked without affecting homeostatic feeding mechanisms.

We conducted a *C. elegans*-based screen of 192 FDA approved or experimental drugs to identify potential adjuvant drugs to pharmacologically suppress AP-induced hyperphagia. Of the four primary hits (doxycycline, minocycline, scopine and vemurafenib) identified, we selected minocycline for further testing in mice due to (i) its ability to selectively block hyperphagia induced by multiple APs (loxapine, olanzapine and chlorpromazine) with only minor effects on basal food intake, (ii) the literature confirming its ability to cross the blood brain barrier,[29] (iii) its excellent safety profile, and (iv) its well described biological effects other than its anti-microbial activity[28].

Follow up studies in mice revealed that minocycline specifically suppressed AP-induced hyperphagia and weight gain while minocycline treatment alone had no effect on food intake or weight gain compared with untreated controls. The effect was not caused by its antibiotic activity, as the structurally closely related tetracycline was unable to suppress AP-induced hyperphagia. Minocycline therefore does not act as a general weight loss drug, but selectively blocks AP-induced hyperphagia. Furthermore, our results show minocycline specifically suppresses the metabolic side effects of olanzapine without affecting its therapeutic action. These data refute previous speculations that weight gain is inextricably linked to the therapeutic effect of APs. Our conclusion is further supported by humans trials in which minocycline was used as an adjuvant to AP treatment for its potential neurocognitive benefits[22,64–66]. The clinical data encouragingly suggest co-administration of minocycline is more likely to improve, rather than reduce, the therapeutic efficacy of APs[67].

A double-blind randomized clinical trial study showed that co-treatment with minocycline significantly reduced weight gain

(4.6 lb vs. 23 lbs in minocycline vs. placebo co-treatment) in response to AP treatment (olanzapine, quetiapine, risperidone or clozapine)[22]. In the AP only control group 100% of patients gained weight while only 40% of the co-treated group gained any weight at all. Two further studies in human patients noted a decrease in AP-induced appetite after minocycline treatment[64] and a trend in reduction in the number of patients gaining a substantial amount of weight[65]. However, a recent study by Liu et al. showed that the combination of minocycline and risperidone was not efficacious in blocking AP-induced weight gain[68]. There are some key differences in these studies that provide a likely explanation for this apparent discrepancy. Liu et al. treated patients who had been established on APs for at least a month while Levkovitz et al. selected patients at the start of AP treatment (less than 14 days). It is important to note that the strongest weight gain effect of APs is often observed in the early phase of treatment[69,70]. For example, after just 12 weeks of treatment 46% of olanzapine treated patients had BMIs in the overweight range compared with just 17.6% at the start of treatment[69]. Similar studies have found AP-induced weight gain occurs in the first year (85% of the total weight gain across treatment), with the biggest leap in the initial 3 months[69]. This suggests the necessity to block the early and dominant effects at the initiation of AP therapy to avoid AP-induced weight gain. In addition, minocycline seems to be most efficacious at blocking olanzapine-induced feeding and less effective at blocking the effects of risperidone, the only AP tested in the study by Liu et al. Our unbiased identification of minocycline, as well as the mechanistic data showing that minocycline specifically blocks AP induced hyperphagic signaling in the hypothalamus strongly supports the positive findings of the Levkovitz et al. study. Our experimental paradigm allows us to test the importance of timing of treatment and efficacy against a broad range of APs as a guide for future clinical trials to study the potential of minocycline and other drugs to abrogate AP-induced hyperphagia and weight gain.

While it is initially surprising that the hyperphagic side effects of APs can be modeled in *C. elegans*, there is no reason to assume that the effect of APs on feeding should be mammalian specific. APs were not designed based on a deep understanding of the changes in neurocircuitry in schizophrenia, but were discovered fortuitously and there is still a general lack of understanding of the specific mechanisms underlying both their therapeutic efficacy and their potent metabolic side effects[49]. Our data suggest that APs act on a core ancestral feeding mechanism present even in *C. elegans*. The finding that *ser-5(ok3087)* mutants or *ser-5(vq1)* knock out animals are resistant to AP-induced hyperphagia, but maintain normal basal food intake and a wild type response to other hyperphagic stimuli indicates the existence of specific regulatory mechanisms controlling hyperphagia that are distinct from basal appetite regulation. Similarly, in mice, minocycline did not act as a general weight loss drug, but specifically blocked AP-induced hyperphagia and weight gain. This specificity was not only observed at the physiological level but also at the level of gene expression where we identified a unique hypothalamic gene expression signature associated with AP-induced hyperphagia that was fully blocked by co-treatment with minocycline. The functional relevance of these genes in AP-induced hyperphagia is supported by the control experiment where we observed that these genes are not differentially expressed in male mice, which are resistant to AP-induced hyperphagia. Furthermore, testing thirteen *C. elegans* mutants for orthologous genes within that signature, revealed that *nfyb-1*/Nfyb and *nhr-25*/Nr5a2 were required for AP-induced hyperphagia. Mutations in either transcription factor blocked AP-induced hyperphagia in *C. elegans*. Further studies in mouse and clinical trials in humans will be needed to test the translational relevance of these findings.

Our unique multi-model approach has identified specific suppressors of AP-induced hyperphagia and weight gain rather than general homeostatic regulators of feeding and body weight regulation. This strategy was made possible by the *C. elegans* based phenotypic screening approach that, without mechanistic knowledge, allowed us to identify adjuvant drugs to block AP-induced hyperphagia. Previous studies in mice[71–74] and humans[75–77] have identified potential adjuvants to block AP-induced weight gain. However, these were selected based on their general anti-obesity properties and thus also have effects on basal appetite regulation. For example, small clinical trials have shown some efficacy of topiramate, samidorphan, exenatide and betahistine but many also induce additional adverse drug reactions and need further study to determine whether they could be of large-scale clinical benefit[76]. However, strategies generally targeting appetite have proven inherently risky due to the central role of basal food intake for organismal health. Our study suggests that it is possible to specifically suppress AP-induced hyperphagia with little to no effect on basal appetite regulation.

## Methods

**Drugs used in *C.elegans* studies.** Chlorprothixene (C1671), flupenthixol (F114), risperidone (R3030), cyproheptadine (C6022), and asenapine (A7861) were ordered from Sigma. Doxycycline (J60579) and serotonin (B21263) were ordered from Alfa Aesar. Haloperidol (153696), minocycline (155718) and chlorpromazine (190326) were ordered from MP Biomedicals. Loxapine (B1001) and quetiapine (B1490) were ordered from ApexBio. Mianserin (0997), olanzapine (4349) and terfenadine (3948) were ordered from Tocris. Ziprasidone (Z0032), scopine (S0912), aripiprazole (A2496), and amitriptyline (A0908) were ordered from TCI. Vemurafenib (V2800) was ordered from LC Laboratories. Lurasidone (RRL001) was ordered from BIOTANG INC. **Drugs used in mouse studies:** Olanzapine (Teva), Minocycline (Aurobindo) and tetracycline-T3383 (Sigma).

***C. elegans* strains.** N2; AQ866 *ser-4(ok512) III*, CX13111 *dop-5(ok568) V*, CF2805 *dop-1(vs100);dop-2(vs105);dop-3(vs106);dop-4(ok1321)*, DA2100 *ser-7(tm1325) X*, DA1814 *ser-1(ok345) X*, RB2277 *ser-5(ok3087) I*, LX734 *dop-2(vs105) V*; *dop-1 (vs100) dop-3(vs106) X*, LX645 *dop-1(vs100) X*, LX636 *dop-1(vs101) X*, LX702 *dop-2 (vs105) V*, FX1062 *dop-2(tm1062)V*, LX703 *dop-3(vs106) X*, BZ873 *dop-3(ok295) X*, FG58 *dop-4(tm1392) X*; RB1680 C24A8.1(ok2090) X, RB1254 C52B11.3(ok1321) X, FX04257 *nfyb-1(tm4257)*, FX04760 *nfyb-1(tm4760)*, OK814 *nfyb-1(cu13) II*, FX03224 *cyp-31A3(tm3224)*, FX01265 Y55F3AM.14(tm1265), FX02908 F42G9.6 (tm2908), FX03026 F42G9.6(tm3026), CB151 *unc-3(e151) X*, JC1971 *tbx-2(ut192) III*, MH1955 *nhr-25(ku217)X*, RB842 *abt-2(ok778)*, RB911 *fshr-1(ok778) V*, IG10 *tol-1(nr2033) I*,VV207 *ser-7(vq2) X*. The VV212 knock out strain was created by injecting sgRNAs flanking the *ser-5* genomic locus (GACGCA-CACCGCCATCTCCAT and TGGGAAGATGAACACGTCCG) together with recombinant Cas9 protein (40μM). Knock outs lacking the *ser-5* gene were identified by PCR. The animals were then outcrossed 4× to eliminate possible off-target effects. The VV207 *ser-7(vq2)* knockout strain was created by injecting sgRNAs flanking the *ser-7* genomic locus (CGTAGACGGGAACATTGCGT and GAAG-CATTATACAACGGGAG) together with recombinant Cas9 protein (40μM). Knockouts lacking the *ser-7* gene were identified by PCR.

**Food intake assay screen in *C. elegans*.** Food intake was measured in N2 (wild-type) *C. elegans* in 96-well microtiter plates using our bacterial clearance assay[23]. Each compound was assayed in 14-21 replicate wells, containing 10–12 animals each. The number of animals per well was counted by eye. The large number of replicates allows for rigorous statistics, captures biological variation, including normality-testing, multiple-hypothesis corrections and false discovery rate determination, as we have done previously[78,79]. Each library (MedChemExpress, #HY-022) compound was initially tested at the highest possible concentration of 33 μM, determined by choosing a safe DMSO concentration (0.3%) that does not interfere with food intake. Each drug was tested in combination with 50 μM loxapine, selected as it is water soluble and thus does not add to the final DMSO concentration. Both library drugs and loxapine were added on day 1 of adulthood, 24 h after sterilization by FUDR (120 μM, Sigma #F0503). Food intake was measured right after adding the drugs and 72 h later, using a 96 well Infinite 200 PRO Tecan plate reader to determine the $OD_{600}$. Compounds that prevent AP-induced food intake were further tested in combination with serotonin (5 mM)[80] to determine whether the primary hit is specific to AP-induced hyperphagia. To avoid indirect effects of the drugs tested in the screen bacteria were killed by γ-irradiation. [https://www.nature.com/protocolexchange/protocols/3245].

**Mouse studies.** All procedures are approved by UCSD or TSRI IACUC committee. *C57BL/6* female mice (stock #000664) were purchased from Jackson labs at

9 weeks of age. All mice were fed normal chow until 10 weeks of age. Olanzapine was compounded into high fat diet (HFD) (45 kcal% from fat) at a concentration of 54 mg/kg of diet (OLZ-HFD, Research Diets, Inc., D09092903, New Brunswick, NJ) as used in other studies[34–36,81,82]. Using the power analysis software and our pilot data we calculated the number of mice needed per group [http://www.statisticalsolutions.net]. Control animals were fed HFD (45 kcal % from fat D09092903). The olanzapine dose selected results in mouse plasma levels (21 ± 5 ng/mL) that are similar to levels observed in humans treated with olanzapine (10–50 ng/mL)[34]. Minocycline (MINO) was administered in the drinking water at a dose of 0.6 mg/mL, which results in a plasma level of less than ~7 μM, concentration comparable with circulating levels in human patients (standard dosing of 50–100 mg minocycline results in plasma level of 2–11 μM[83,84]. At 10 weeks of age mice were randomly divided into four treatment groups (control, OLZ, MINO and OLZ + MINO) and fed these diet/water combinations for the 12 week duration of the study. At 11 weeks of treatment mice were placed in EchoMRI machine to measure fat and lean mass and then placed in Comprehensive Lab Animal Monitoring System (CLAMS) for 3 days to acclimate followed by 2 days of recordings to measure oxygen consumption ($\dot{V}O_2$), carbon dioxide production ($\dot{V}CO_2$), respiratory exchange ratio (RER), energy expenditure, total activity at $x$-axis (XTOT) and total activity at $z$-axis (ZTOT). At 12 weeks mice were sacrificed and samples snap frozen in liquid nitrogen. Circulating levels of cytokines and gut-derived hormones were measured by multiplex ELISA (lincoplex, st Charles, MO). As a control experiment male C57BL/6 mice (stock #000664) purchased from Jackson labs, were fed either olanzapine-HFD or control HFD from 12 weeks of age for 2 weeks ($n = 7$ per group). At 14 weeks of age mice were sacrificed and samples snap frozen in liquid nitrogen. Tetracycline co-treatment study: 10 week old female mice were divided into 6 groups (control, OLZ, MINO, MINO + OLZ, TETRA and TETRA + OLZ, $n = 6–8$ per group). Mice were treated with olanzapine and minocycline as described above, while additional two groups were treated with tetracycline (0.6 mg/ml in drinking water) either alone (TETRA) or in combination with olanzapine (TETRA + OLZ). Food intake and weight gain were measured daily throughout the two-week study. Amphetamine-induced hyper locomotion assay[38]. Twelve week old female C57BL/6 mice were divided into 4 groups (control, OLZ, MINO and MINO + OLZ, $n = 12$ per group). MINO was provided through the drinking water (0.6 mg/mL). One week into these treatments, the mice were habituated to the locomotor activity test (1 hr) following injection of saline (0.9%, IP, 0.01 mL/g), which was repeated 3 days later. Three days after the second habituation, OLZ (2 mg/kg, 0.01 mL/g, in 1% methylcellulose) or vehicle (1% methylcellulose) was injected IP and the mice were immediately placed in the locomotor activity chambers. 30 min later the mice were removed very briefly from the chamber to receive 2.5 mg/kg D-amphetamine (in 0.9% saline, 0.01 mL/g, IP) and replaced in the system for an additional 120 min. Locomotor activity was measured in polycarbonate cages ($42 \times 22 \times 20$ cm) placed into frames ($25.5 \times 47$ cm) mounted with two levels of photocell beams at 2 and 7 cm above the bottom of the cage (San Diego Instruments, San Diego, CA) to record both horizontal (ambulation, center activity and total horizontal activity) and vertical (rearing) behavior every minute for the 30 min pre-amphetamine and 120 min post-amphetamine test phase. Data were analyzed using repeated measures ANOVA followed by Tukey's multiple comparisons test.

**RNA extraction and sequencing**. RNA extraction and sequencing was performed as previously described[85–87]. Briefly, total RNA was isolated using TRIzol (Invitrogen) according to the manufacturer's instructions. The quality of the RNA was assessed using the Tapestation 2200 (Agilent) and Libraries prepared using TruSeq Library prep kits (Illumina), and then run on the Tapestation high sensitivity DNA assay kits to ensure correct library size. Libraries were quantified using the Qubit® 2.0 Fluorometer, pooled and run on the Hiseq 2500 (Illumina). Reads were mapped to the mouse transcriptome using Bowtie2 algorithm[88] and counted as reads per gene using RSEM[89] and then analyzed using the statistical algorithm limma. Genes were then sorted by their posterior error probability (local false discovery rate, FDR). We also express significance of a gene as the $q$ value, which is the smallest false discovery rate at which a gene is deemed differentially expressed. All RNA-seq data can be found at Gene Expression Omnibus (GEO) database, [http://www.ncbi.nlm.nih.gov/geo] (accession GSE119772).

**Statistics**. Due the characteristics of the different data sets a series of statistical tests and representations were used as indicated in the figure legends. For measuring food intake in response to a single dose antipsychotic we used Tukey style plots to represent the large variance in food intake data that is due biological origin. For EC50 plots we used S.E.M. of 14–21 population of 5–15 worms each. Comparing the effect of different conditions we either used $t$-test or ANOVA followed by a Bonferroni correction, dependent on the number of comparisons. For each condition we measured food intake of an average of 426 worms except for the screen, where we only measured 96 animals per drug. In total we determined food intake of 161,102 worms.

**Data availability**
The authors declare that all data supporting the findings of this study are available within the paper (and its Supplementary Information files), at Gene Expression

Omnibus (GEO) database, (accession GSE119772) or available upon request from the corresponding authors. The Source Data for all figures are provided as a Source Data file.

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

## Acknowledgements
These studies were supported by NIH grant R01DK117872 awarded to M.P (TRSI) and O.O. (UCSD). Additional support was provided by grants NIH grant UL1TR001442 (O. O.) and DP2OD008398 (M.P.). We would like to thank Song Qu for technical help and Meng Wang's laboratory for providing strains. Some strains were also provided by the CGC, which is funded by NIH Office of Research Infrastructure Programs (P40 OD010440). We would like to thank Dr. Amanda Roberts for her assistance with the amphetamine-induced hyperlocomotion experiments and Steve Head from the sequencing facility for help with the RNA seq. We would like to thank William Ja (TSRI Florida) and Katharina Brandl (UCSD, School of Pharmacy) for comments on the manuscript and Kristin Cadenhead (UCSD, Department of Psychiatry) for advice and for providing a clinical perspective regarding our findings.

## Author contributions
O.O. and M.P. conceived, designed the studies and wrote the manuscript. A.P.-G. and M. C.-R. also designed the studies, performed the experiments and analyzed data. N.W., V. P., A.T., V.T., and G.S. performed experiments and analyzed data.

## Additional information

**Competing interests:** The authors declare no competing interests.

