## [Peer Review File · Nature Communications]

Reviewers' comments:

Reviewer #1 (Remarks to the Author):

Perez-Gomez and colleagues describe an elegant screen to identify small molecules that target feeding mechanisms that are inappropriately activated by antipsychotics. The premise of the work is clear and compellingly articulated. The authors alternately use a *C. elegans* model for screening and analysis of candidate genes and a rodent model to determine whether drug-effects observed in nematodes are also observed in mammals. Specifically, they demonstrate that a class of antipsychotics increases feeding by *C. elegans* as it does in mammals. After demonstrating that antipsychotics increase feeding through a genetic pathway that is distinct from the pathway controlling basal feeding, they screen a small library of FDA-approved compounds for drugs that have little or no effect on basal feeding but block antipsychotic-induced feeding. This screen yields the tetracycline analog minocycline, which the authors find also blocks antipsychotic-activated feeding in a mouse model and suppresses a characteristic transcriptional response to antipsychotics in the rodent hypothalamus. The authors finally return to the nematode model to investigate two genes that are regulated by antipsychotics and report that a nuclear hormone receptor and the transcription factor *Nfyb* are involved in the actions of antipsychotics on feeding.

Strengths

1. The authors use a rigorous and novel assay to measure drug-effects on *C. elegans* feeding. Their approach is well suited to screening, as shown by this work. Data are clearly presented, and the authors take care to explain analysis and statistical tests used to make their conclusions.
2. A role for a gene in drug-effects on feeding is often corroborated by analysis of two independently derived mutant strains.
3. A two-tiered screening strategy that uses *C. elegans* and rodents is ambitious and satisfying. It is remarkable to see a candidate molecule carried across the finish line.
4. The authors not only assess effects of a drug on feeding but also look for transcriptional changes in hypothalamus. These analyses are independent of each other and reinforce the authors' conclusion that minocycline exerts specific effects on targets that are inappropriately activated by antipsychotics.
5. The manuscript is written in a clear and frank manner. This is especially evident in the discussion, which summarizes data from clinical trials.
6. The authors' initial analysis of the roles of different monoamine receptors in basal and antipsychotic-enhanced feeding reveals some interesting and novel biology. These data are a foundation for the screen that is the subject of this work, but they are also a gateway to new molecular genetic studies of the control of feeding by *C. elegans*.

Weaknesses

1. The authors' discussion of the *ser-7* phenotype shown in Fig. 1E does not seem to reflect what is shown in the data. I see an effect of *ser-7* mutation on responses to antipsychotic while the text emphasizes that there is still a statistically significant difference between controls and antipsychotic-treated. This is minor.
2. Because of point 1, I am interested to know if *ser-5*; *ser-7* double mutants are completely defective in serotonin and loxapine-evoked feeding. This is a straightforward experiment that might nicely account for the effects of these compounds on feeding. The absence of these data does not greatly

diminish the impact of the story, so I would also rate this as a minor issue but one that I would encourage the authors to consider.

3. The authors guide the reader through the logic that lead them to minocycline. A key question that they ask is whether minocycline exerts its effects on *C. elegans* feeding through its antibiotic activity. They show that this is not likely the case because other antibiotics do not affect antipsychotic-induced feeding. The same logic is not followed for studies of mice. The authors should test at least one related antibiotic to determine whether minocycline effects on rodent feeding, like its effects on *C. elegans* feeding, can be uncoupled from its action as an antibiotic. I rate this as an important matter to be addressed.

4. The rationale for testing whether *nhr-25* and *nfyb-1* are required for effects of antipsychotics on feeding is that their homologs are upregulated in hypothalamus upon exposure to antipsychotics. The authors should test whether antipsychotics similarly alter expression of these genes in nematodes. It is possible that straightforward ways to tackle this question will not yield clear answers, for example if gene expression is regulated only in specific cells. The authors should nevertheless attempt to answer this question to determine the extent of conservation between mouse and worm pathways. I rate this as a moderately important matter; the experiments should be done because a positive result would advance the narrative, but a negative result would not diminish the claims of the paper.

5. I could only find analysis of a single allele of *nhr-25*. The authors should confirm this phenotype with a second allele.

Minor comments

1. Conform to standard nomenclature for alleles e.g. *ser-5(vq1)*

2. The authors' description of clinical trials that have already shown effects of minocycline on patients taking antipsychotics is frank and helpful but also comes out of the blue in this paper's narrative. It reads as if after all of this screening, the authors have found a compound that was already suggested or known to have these desirable properties. It is clear to me that the authors' work provides a substantial foundation for dissecting the effects of minocycline and related compounds on feeding using a *C. elegans* model. To avoid any confusion about the contribution that this work makes to the field, the authors should emphasize that their approach was discovery-based and hypothesis-free and that for the first time there are molecular pathways associated with these previously cryptic effects of minocycline.

Reviewer #2 (Remarks to the Author):

Review of 'A Phenotypic *C. elegans* Screen Identifies a Selective Suppressor of Antipsychotic-induced Hyperphagia' by Perez-Gomez et al. for Nature Communications

It was with much interest that I read Perez-Gomez's work on identifying therapeutic candidates to treat weight gain associated with anti-psychotic use using *C. elegans* as a primary model system. With a couple of exceptions as noted below, I find the story well-written and the data compelling, beautiful, and well-communicated.

One question that I kept asking myself throughout the manuscript, including in the abstract, is whether the hits are simply suppressors of the anti-psychotic, and therefore not only suppress weight gain, but also diminish the psychological therapeutic effect of the primary drug. This should be

addressed.

A second issue that I actually felt betrayed by (in a literary sense) is that the authors only reveal in the middle of the discussion that it has already been shown that one of their top hits has been shown to be clinically useful to reduce the weight gain associated with AP. I read the manuscript with much enthusiasm with hope building that the authors were on the verge of discovering a much needed way of reducing AP side-effects only to learn that someone beat them to the chase eight years ago. Realizing this, it forced me to re-read parts of the manuscript to learn how I could have been misled (unintentionally, I'm certain).

I think the knowledge that minocycline suppresses the weight-gain of at least one AP has been presented earlier in the manuscript, and that the story should be reframed to include that. Something along the lines of building a better pipeline to identify more adjuvant options, or simple model systems that can be used to understand the MOA of effective adjuvants like minocycline.

Minor issues:

1. Figure 2B- it wasn't clear which dot is scoline, and why dox and vemurafenib are not indicated.
2. The authors present a beautiful vignette about the orexigenic peptides Npy and AgRP in mice, but when they return to elegans, they make no mention of them. I assume that conservation is non-existent or at least undetectable. Still, given how the story is told, the presence/absence/ambiguity of the two genes in elegans should be explicitly addressed.

Response to Reviewers comments:

We wish to thank the reviewers for their comments and their constructive criticism. We believe answering them has significantly improved the manuscript. We address their concerns point by point in the response below:

Reviewer #1 : Weaknesses:

1. The authors' discussion of the *ser-7* phenotype shown in Fig. 1E does not seem to reflect what is shown in the data. I see an effect of *ser-7* mutation on responses to antipsychotic while the text emphasizes that there is still a statistically significant difference between controls and antipsychotic-treated. This is minor.

Answer: Reviewer #1 is correct in pointing out that the single dose experiment shown in Figure 1E, reflects a mild increase in food intake of the *ser-7(tm1325)* in response to loxapine. To address this concern we generated a complete *ser-7* knock out using Crisper / Cas9 *ser-7(vq2)* and measured dose response curves. These new data clearly show that *ser-7(vq2)* animals elicit a hyperphagic response to olanzapine (new figure: Fig. 1G) but do not respond to serotonin.

See lines 134-150 and new figure 1G.

2. Because of point 1, I am interested to know if *ser-5*; *ser-7* double mutants are completely defective in serotonin and loxapine-evoked feeding. This is a straightforward experiment that might nicely account for the effects of these compounds on feeding. The absence of these data does not greatly diminish the impact of the story, so I would also rate this as a minor issue but one that I would encourage the authors to consider.

Answer: To address this point we used Crispr/Cas9 to generate individual knockout strains (*ser-5(vq1)* and *ser-7(vq2)*) and then crossed them together produce a double knockout strain. As predicted, the double mutant does not elicit a hyperphagic response to either olanzapine or serotonin (new figure: Fig. 1H). However, in contrast to the single mutants the double mutant shows a clear reduction in basal feeding. These animals appeared “healthy” suggesting the reduction in feeding is unlikely to be due to general “sickness” of the animals.

See lines 134-150 and figure 1H.

H

3. The authors guide the reader through the logic that lead them to minocycline. A key question that they ask is whether minocycline exerts its effects on *C. elegans* feeding through its antibiotic activity. They show that this is not likely the case because other antibiotics do not affect antipsychotic-induced feeding. The same logic is not followed for studies of mice. The authors should test at least one related antibiotic to determine whether minocycline effects on rodent feeding, like its effects on *C. elegans* feeding, can be uncoupled from its action as an antibiotic. I rate this as an important matter to be addressed.

Answer: While we had conducted studies testing closely related antibiotics in worms for their potential to abrogate AP-induced food intake we had not verified this in mice. We have now conducted a study testing the effect of Tetracycline on AP-induced food intake (see supplemental figure 2, also included below for ease of reference). Mice were treated for 2 weeks with tetracycline (TETRA) in the drinking water at the same concentration used for minocycline (MINO, 0.6mg/ml). Tetracycline treatment alone had no effect on food intake or weight gain. Co-treatment of tetracycline with olanzapine (TETRA+OLZ) did not blunt olanzapine-induced food intake or weight gain. Therefore, we conclude that minocycline suppression of AP-induced food intake and weight gain is via a mechanism distinct from its antibiotic properties as closely related antibiotics do not suppress AP-induced food intake or weight gain.

These new data have now been added to the manuscript, see supplemental figure 2 and manuscript lines 212-218, 443-460 and 565-569.

4. The rationale for testing whether *nhr-25* and *nfyb-1* are required for effects of antipsychotics on feeding is that their homologs are up-regulated in hypothalamus upon exposure to antipsychotics. The authors should test whether antipsychotics similarly alter expression of these genes in nematodes. It is possible that straightforward ways to tackle this question will not yield clear answers, for example if gene expression is regulated only in specific cells. The authors should nevertheless attempt to answer this question to determine the extent of conservation between mouse and worm pathways. I rate this as a moderately important matter; the experiments should be done because a positive result would advance the narrative, but a negative result would not diminish the claims of the paper.

Answer: We conducted qPCR experiments for *nhr-25* and *nfyb-1* using RNA from the whole animals after AP treatment. We did not observe any significant changes in expression of either gene (data not shown). We hypothesize that any expression changes may occur in distinct tissues or neuronal populations and we are unable to detect these differences when preparing RNA from the whole animal.

5. I could only find analysis of a single allele of *nhr-25*. The authors should confirm this phenotype with a second allele.

Answer: Unfortunately, there is only one viable mutant strain of *Nhr-25* available.

The *nhr-25(ku215)* allele we used is a temperature sensitive mutant. We have attempted to conduct a food intake experiment at 15°C, a temperature at which the strain is effectively “wild type”. However, changing the temperature has a dramatic effect on food intake even on wild type N2 worms with the result that we are no longer able to reliably measure changes in food intake in response to antipsychotics.

We did not attempt to use RNAi as we have yet to establish a protocol to combine the use of RNAi with our feeding assay and antipsychotics. The RNAi feeding library

requires us to use a different E.coli strain (HT115) which we have not yet optimized in our system. In addition live bacteria need to be used and in our experience treatment with APs can effect bacterial growth and thus is a confounding variable that we have thus far eliminated by using irradiated bacteria in our C. elegans screen.

Minor comments

1. Conform to standard nomenclature for alleles e.g. ser-5(vq1)

Answer: We have now removed the delta sign in front of the knock out strains and just indicated the allele vq1 to conform with standard nomenclature.

2. The authors' description of clinical trials that have already shown effects of minocycline on patients taking antipsychotics is frank and helpful but also comes out of the blue in this paper's narrative. It reads as if after all of this screening, the authors have found a compound that was already suggested or known to have these desirable properties. It is clear to me that the authors' work provides a substantial foundation for dissecting the effects of minocycline and related compounds on feeding using a C. elegans model. To avoid any confusion about the contribution that this work makes to the field, the authors should emphasize that their approach was discovery-based and hypothesis-free and that for the first time there are molecular pathways associated with these previously cryptic effects of minocycline.

Answer: We now mention these clinical findings up front in the introduction, (lines 95-100).

“Minocycline has also shown efficacy in reducing weight gain in response to AP treatment in clinical studies where only 40% of the co-treated group gained any weight at all compared with 100% in the control group⁶⁰. Our discovery based and hypothesis-free C. elegans screening strategy facilitates the identification of other potential adjuvant options as well as gain deeper mechanistic understanding in to the mode of action of effective adjuvants like minocycline”.

We then emphasize and further discuss the importance of these clinical studies in lines 327-348.

Reviewer #2 (Remarks to the Author):

1. One question that I kept asking myself throughout the manuscript, including in the abstract, is whether the hits are simply suppressors of the anti-psychotic, and therefore not only suppress weight gain, but also diminish the psychological therapeutic effect of the primary drug. This should be addressed.

Answer: This is a very valid concern especially in the light of previous speculation that weight gain effect of APs is the consequence of an on-target effect that cannot be

separated from its therapeutic effect. We have addressed this concern experimentally (new figure: Fig. 3l). We used the amphetamine-induced hyperlocomotion model, a standard test to determine antipsychotic efficacy in rodents. In this assay an amphetamine injection induces a dopamine burst that induces hyperlocomotion that is blocked by antipsychotics such as olanzapine. To test the effect of minocycline on the efficacy of olanzapine, we treated female C57BL7J mice (=12/ cohort, 4 groups) for 2 weeks with minocycline in the drinking water (0.6mg/ml). As can be seen in the new Fig. 3l olanzapine blunts the hyperlocomotion induced by amphetamine. Minocycline treatment has no effect on the ability of olanzapine to blunt amphetamine-induced hyperlocomotion. Thus minocycline specifically blocks the metabolic side-effects of olanzapine without affecting the therapeutic efficacy of olanzapine.

See lines 228-234 and new figure 3l.

Co-incidentally minocycline has been tested as an add-on therapy to AP- treatment and resulted in some improvement in the negative cognitive symptoms associated with schizophrenia. This strongly suggests that minocycline does not diminish the beneficial psychiatric effect of APs but if anything improves symptoms that current APs do not effectively manage. (Chaudhry, 2012; Kelly, 2015; Khodaie-Ardakani, 2014; Levkovitz, 2010; Liu, 2014; Xiang, 2017).

See lines 315-325

2. A second issue that I actually felt betrayed by (in a literary sense) is that the authors only reveal in the middle of the discussion that it has already been shown that one of their top hits has been shown to be clinically useful to reduce the weight gain associated with AP. I read the manuscript with much enthusiasm with hope building that the authors were on the verge of discovering a much needed way of reducing AP side-effects only to learn that someone beat them to the chase eight years ago. Realizing this, it forced me to re-read parts of the manuscript to learn how I could have been misled (unintentionally, I'm certain). I think the knowledge that minocycline suppresses the weight-gain of at least one AP has been presented earlier in the manuscript, and that the story should be reframed to include that. Something along the lines of building a better

pipeline to identify more adjuvant options, or simple model systems that can be used to understand the MOA of effective adjuvants like minocycline.

Answer: It was certainly not our intention to mislead the reviewer. To make it clear upfront that the clinical data exists we now raise this point in the introduction (lines 95-100).

“Minocycline has also shown efficacy in reducing weight gain in response to AP treatment in clinical studies where only 40% of the co-treated group gained any weight at all compared with 100% in the control group⁶⁰. Our discovery based and hypothesis-free C. elegans screening strategy facilitates the identification of other potential adjuvant options as well as gain deeper mechanistic understanding in to the mode of action of effective adjuvants like minocycline”.

We then emphasize and further discuss the importance of these clinical studies in lines 327-348.

Minor issues:

1. Figure 2B- it wasn't clear which dot is scopine, and why dox and vemurafenib are not indicated.

Answer: We now have colored all hits that scored positive in red in Fig. 2.

2. The authors present a beautiful vignette about the orexigenic peptides Npy and AgRP in mice, but when they return to C. elegans, they make no mention of them. I assume that conservation is non-existent or at least undetectable. Still, given how the story is told, the presence/absence/ambiguity of the two genes in elegans should be explicitly addressed.

Answer: The C. elegans genome contains 113 neuropeptide genes that code for 250 different neuropeptides but the *clear orthologs of Npy or Agrp have not been described in C.elegans*. We have now added this to the manuscript (lines 273-275).

“Importantly, C.elegans has many conserved neuropeptide signaling systems that regulate metabolic pathways involved in appetite, satiety, energy homeostasis and fat metabolism⁴⁶ but clear orthologs of Npy or Agrp have not been described in C.elegans⁴⁷⁻⁴⁹. “

REVIEWERS' COMMENTS:

Reviewer #1 (Remarks to the Author):

Perez-Gomez and colleagues have revised their manuscript by including significant new data and revisions that clarify and expand their arguments. They have thoroughly addressed substantive questions raised during review and provided satisfactory responses to matters that could not be readily addressed. The manuscript has been strengthened, and I have no further questions or concerns.

Reviewer #2 (Remarks to the Author):

The authors went through significant effort to satisfactorily address my comments (and looks like they did a thorough job addressing the other reviewer's comments). I am happy with the changes and support the manuscript's publication.